# High-speed scanless entire bandwidth mid-infrared chemical imaging

Yue Zhao [1,5] ✉, Shota Kusama[1], Yuji Furutani [2,3], Wei-Hong Huang[4], Chih-Wei Luo [4] & Takao Fuji [1] ✉

Mid-infrared spectroscopy probes molecular vibrations to identify chemical species and functional groups. Therefore, mid-infrared hyperspectral imaging is one of the most powerful and promising candidates for chemical imaging using optical methods. Yet high-speed and entire bandwidth mid-infrared hyperspectral imaging has not been realized. Here we report a mid-infrared hyperspectral chemical imaging technique that uses chirped pulse upconversion of sub-cycle pulses at the image plane. This technique offers a lateral resolution of 15 μm, and the field of view is adjustable between 800 μm × 600 μm to 12 mm × 9 mm. The hyperspectral imaging produces a 640 × 480 pixel image in 8 s, which covers a spectral range of 640–3015 cm$^{-1}$, comprising 1069 wavelength points and offering a wavenumber resolution of 2.6–3.7 cm$^{-1}$. For discrete frequency mid-infrared imaging, the measurement speed reaches a frame rate of 5 kHz, the repetition rate of the laser. As a demonstration, we effectively identified and mapped different components in a microfluidic device, plant cell, and mouse embryo section. The great capacity and latent force of this technique in chemical imaging promise to be applied to many fields such as chemical analysis, biology, and medicine.

Hyperspectral imaging is a mapping technique that combines imaging and spectroscopy to characterize and identify the distribution of chemical constituents. A spectral image is a data cube with two spatial dimensions and one spectral dimension, namely, the full spectrum can be extracted from each pixel in the hyperspectral image. A general color image based on the RGB color model has only 3 bands and a multiband with more than 3 bands (generally less than 10) is called a multispectral technique, while hyperspectral imaging has hundreds to thousands of bands. In recent years, hyperspectral images in the visible and near-infrared band have been widely used in many fields, such as food[1,2], medical[3,4], pharmaceutical[5,6], microbial[7,8], nanoscience[9–12], agriculture[13–18], forestry[19], marine science[20–22], environmental monitoring[23,24], geology[25,26], ecology[27], forensic medicine[28,29], and lunar

and planetary surface science[30,31]. The mid-infrared (MIR) hyperspectral technique enables chemical imaging or chemical mapping by providing information on the chemical composition of an object through molecular vibrations. In the early period, researchers and technicians tried to use the well-established FT-IR technology for chemical imaging[32–35]. For mid-infrared light, a mercury–cadmium–telluride (MCT) detector is generally used for photothermal imaging. However, since MCT detectors are subject to thermally excited electronic noise, even with cryogenic cooling to suppress this noise, the signal–to–noise ratio (SNR) is much lower than that of Si-based detectors.

Another limitation of MIR imaging is the light source. For example, although the incoherent IR light source emits a continuous spectrum of 1–25 μm, the intensity cannot satisfy the SNR required for

[1]Laser Science Laboratory, Toyota Technological Institute, 2–12–1 Hisakata, Tempaku-ku, Nagoya 468-8511, Japan. [2]Department of Life Science and Applied Chemistry, Nagoya Institute of Technology, Showa-Ku, Nagoya 466-8555, Japan. [3]Optobiotechnology Research Center, Nagoya Institute of Technology, Showa-Ku, Nagoya 466-8555, Japan. [4]Department of Electrophysics, National Yang Ming Chiao Tung University, Hsinchu 30010, Taiwan. [5]Present address: Graduate School of Engineering College of Design and Manufacturing Technology, Muroran Institute of Technology, 27-1 Mizumoto-cho, Muroran, Hokkaido 050-8585, Japan. ✉e-mail: zhaoyue@muroran-it.ac.jp; fuji@toyota-ti.ac.jp

scanless imaging with MCT cameras. In recent years, wavelength-tunable quantum cascade lasers (QCL) have been used as a light source for MIR imaging with a useful SNR range for MCT detectors[36,37]. However, the disadvantage of QCL is the limited bandwidth. Due to the limitation of the wavelength, a few wavelength points were used to identify substances. For example, 5 kinds of mono- and disaccharides can be classified by picking 10 distinct wavenumbers[38]. As an alternative, 4 separate QCL tunable ranges are required to continuously cover the 900–1800 cm$^{-1}$ [39] or 770–1940 cm$^{-1}$ [40,41]. wavenumber range. Therefore, QCL-based MIR imaging is currently more suitable for discrete frequency imaging. To our knowledge, the highest frame rate of QCL-based imaging at a single wavenumber reached 50 Hz in 640 × 480 pixels[37], while 480 × 480 pixel imaging with three discrete frequencies required 54 s/frame, and a wide wavenumber imaging at 900–1800 cm$^{-1}$ required at least 5 min/frame[42]. The measurement speed would be improved by using an optical parametric amplifier (OPA) which is in the wavenumber range of 1162–1562 cm$^{-1}$ and 2020–2500 cm$^{-1}$ [43]. However, full-spectrum MIR imaging still presents technical challenges due to the limitation of the output wavelength bandwidth of OPA.

Another strategy is up-conversion of the MIR pulses to visible or near-infrared light. Visible or near-infrared light can be detected with Si-based detectors, which have much higher performance than MIR detectors[44,45]. In the demonstrated hyperspectral imaging based on the up-conversion method, the length of the crystal should be ~10 mm to have a reasonable conversion efficiency. In general, such a long crystal has a narrow phase matching bandwidth. Therefore, the crystal needs to be rotated to scan the wavelength, and each measured frame is post-processed for image reconstruction[44,45]. More recently, higher nonlinear efficiencies have been implemented using metasurfaces instead of nonlinear crystals[46]. However, the wavelengths generated by OPA are difficult to generate light in the fingerprint region (<1500 cm$^{-1}$). Although the above methods have shown many excellent imaging results, especially applied in the imaging of living tissue, the imaging speed and the wavenumber bandwidth are not enough to meet the requirements of high-speed chemical imaging. Concerning the wavenumber bandwidth, the fingerprint region (<1500 cm$^{-1}$) is generally preferred in qualitative chemical species, but the functional group region (≥1500 cm$^{-1}$) can be used to determine the double, triple, X−H bonds, and functional group. Therefore, simultaneous high-speed imaging of the fingerprint region and the functional group region is crucial.

Here, we propose the most efficient mid-infrared hyperspectral chemical imaging method to our knowledge, which employs scanless global imaging to save time on raster scans. Additionally, it covers the entire MIR bandwidth, and we can create either a monochromatic or a hyperspectral image by using a wavelength filter or a grating to save time on tuning wavelengths.

A general method for characterizing ultrashort mid-infrared pulses is through upconversion using a stretched chirped pulse[47–50]. We used self-developed, full-spectrum (bandwidth: 3−30 μm), sub-cycle MIR pulses as the light source and performed global imaging, which can produce an image without the need for sample scanning. The MIR image of the sample was then projected onto a nonlinear crystal. The MIR signals were converted to visible light while preserving absorption spectrum information. Since the pulse energy of the light source was sufficient for full wavelength upconversion efficiency, a thin crystal (<5 μm) with a wide phase matching bandwidth can be used. Finally, hyperspectral images were obtained by a hyperspectral camera with a silicon-based detector. The innovation of the MIR light source and the fusion with the snapshot method and upconversion not only meets the requirements of the spectrum for identifying chemical substances in chemical imaging but also dramatically improves the speed of chemical imaging.

## Results

### Experimental setup

As shown in the conceptual diagram of Fig. 1a, the whole sample is excited with a global irradiation by the mid-infrared pulses $E_{MIR}(t)$ with a pulse width of 13.6 fs which is sub-cycle phase-stable pulses and a spectral bandwidth of 3–30 μm[51]. After the sample, a chirped pulse $E_{CP}(t)$ with a duration of 1.8 ps and a center wavelength of 790 nm is introduced and is coaxialized with the transmitted MIR beam. A nonlinear crystal GaSe is placed on the image plane, and the MIR pulse $E_{MIR}(t)$ is converted to visible light $E_{CP}(t)E_{MIR}(t)$. Since the chirped pulse is used for the upconversion, the MIR pulse with a tail of picosecond free induction decay (FID) signals is fully converted to visible, and the information of the MIR spectrum is maintained with the resolution of the inverse of the chirped pulse duration. The upconverted light is detected with a hyperspectral camera, and a hyperspectral image, namely the spectral image data cube, is obtained. Finally, the components of the sample are mapped by spectral analysis.

The experimental setup of the MIR hyperspectral imaging system is shown in Fig. 1b. We should note that the peak power of the sub-cycle pulses is much higher than that of other existing MIR light sources, thus even though the beam diameter of the MIR was close to 3 mm on the image plane, the up-converted light intensity was still sufficient to meet the SNR required for hyperspectral imaging. In any other MIR hyperspectral imaging techniques based on upconversion, the MIR light is upconverted at the Fourier plane. Although the highest intensity of the MIR light and the largest upconversion efficiency are expected at the Fourier plane, the largest issue is that each size of the image at each wavelength becomes different and one has to compensate for it by considering the complex phase matching conditions. On the other hand, the MIR light is upconverted at the image plane in our case, thus each size of the image at each wavelength becomes identical. This is one of the largest benefits of our imaging concept.

The light source is a Ti:sapphire laser system based on a regenerative amplifier and a single-pass amplifier (Spectra-Physics Spitfire Ace) with a pulse duration of 35 fs, output pulse energy of 2.8 mJ, repetition rate of 5 kHz, and wavelength of 800 nm ($\omega_F$). The laser beam was split into two by a beam splitter with transmittance and reflectance of 7:3. The transmitted beam was used for the generation of the MIR pulse through two-color filamentation[51] and the reflected beam was used for the chirped pulse.

The transmitted beam was sent to a doubling crystal (β-BBO, type-I, cut angle 29˚, thickness 100 μm) to generate a second harmonic (SH) pulse ($\omega_{SHG}$, 406 nm). The collinear two-color pulses pass through a birefringent calcite crystal (thickness 1.7 mm) and the relative delay between them is aligned by controlling the angle of the crystal. After that, a dual wave plate ($\lambda$ for $\omega_{SHG}$, $\lambda/2$ for $\omega_F$) is used for rotating the polarization of the fundamental pulse so that the fundamental and SH pulses have the same polarization and the efficiency of the four-wave difference frequency generation (FW-DFG) process is optimized. The fundamental and SH beams were focused by a concave mirror (CM1, highly reflects 800 nm and 400 nm wavelength, radius of curvature: 1.5 m). The MIR pulse (pulse width 13.6 fs, pulse energy 1.2 μJ) was generated through the two-color filamentation. After the filament, the beam is collimated with an aluminum-coated concave mirror (CM2, radius of curvature: 0.5 m) with a 7 mm diameter hole. Since the generated MIR beam has a much larger divergence than the excitation beam, the fundamental and SH beams are removed by 83.5% in this way. The residual fundamental and SH beams were reflected and then removed by a 200 μm thick silicon wafer. The silicon wafer was set at Brewster's angle to maximize the transmittance of the MIR beam. After that, the MIR beam was focused and irradiated to the sample by a concave mirror (CM3, gold coating, radius of curvature: 0.5 m). The image of the sample was transferred to the sum frequency generation (SFG) crystal by the two concave mirrors (CM4 and CM5, gold coating, radius of curvature: 0.4 m and 0.3 m respectively). Assuming the

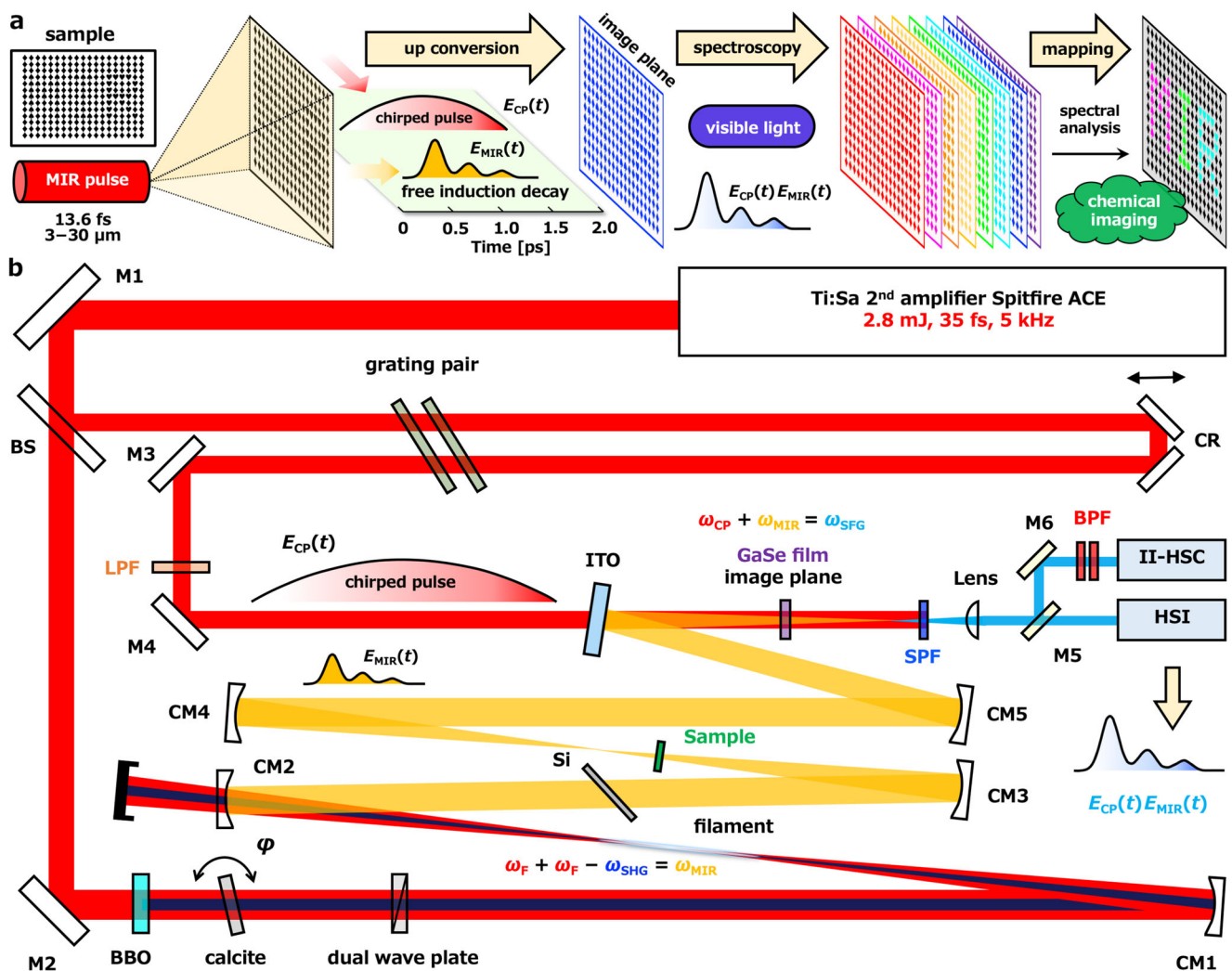

**Fig. 1 | Concept and setup of the high-speed scanless mid-infrared hyperspectral chemical imaging. a** Conceptual illustration of high-speed scanless mid-infrared (MIR) hyperspectral chemical imaging. **b** Mid-infrared hyperspectral chemical imaging setup. M1, M2, M3, and M4: mirrors (high reflection for $\omega_F$), BS: beam splitter with 70% transmittance and 30% reflectance. BBO: β-BaB$_2$O$_4$ (type-I, cut angle 29°, thickness 100 μm). CM1: dielectric concave mirror (high reflection for $\omega_F$ and $\omega_{SHG}$, radius of curvature: 1.5 m), CM2: aluminium-coated concave mirror (radius of curvature: 0.5 m) with a hole of 7 mm diameter, CM3, CM4, and CM5: gold-coated concave mirrors (radius of curvature, CM3: 0.5 m, CM4: 0.4 m and CM5: 0.3 m), CR: corner reflector with two mirrors (high reflection for $\omega_F$), LPF: long pass filter (Semrock FF01-776/LP-25, cut-on wavelength: 783 nm), SPF: short pass filter (Semrock FF01-770/SP-25, cut-on wavelength: 752 nm), Lens: achromatic lens (focal length 200 mm), M5 and M6: aluminium-coated mirrors, BPF: tunerable bandpass filter set (Semrock TBP01-704/13-25x36 and TBP01-790/12-25x36), HSI: hyperspectral imaging unit (EBA JAPAN, SIS-H-0.45nm, number of effective pixels: 640 × 480), II-HSC: a high-speed gated image intensifier unit (Hamamatsu, 10880-13F) and a high-speed camera (Phantom, VEO 410L).

optical system from the sample to the sum frequency generation crystal plane as a microscope, the combination of CM4 and CM5 acts as an objective lens. A ~4.4 μm thick GaSe crystal film (cut angle 90°) was set on the image plane and the crystal surface was parallel to the image plane.

The reflected beam from the beam splitter was stretched to a chirped pulse of 1.8 ps by using a pair of diffraction gratings. The chirped pulse was overlapped with the MIR pulse on the ITO mirror. The MIR pulse $E_{MIR}(t)$ and the chirped pulse $E_{CP}(t)$ are sent to the GaSe crystal film and the sum frequency between the chirped pulse and the MIR pulse is generated. The phase matching condition is Type I, namely, sum frequency (extraordinary wave) generation between the chirped pulse (ordinary wave) and MIR (ordinary wave). The SFG signal $E_{CP}(t)E_{MIR}(t)$ preserves the information of the MIR spectrum. The long wavelength component of SFG, which corresponds to the low-frequency component of MIR, overlaps with the short wavelength component of the chirped pulse itself. Therefore, a long pass filter (cut off at 783 nm) on the chirped pulse and a short

pass filter (cut off at 752 nm) on the SFG were set to avoid wavelength overlap. As a result, the measurable wavelengths were limited to 638–752 nm which corresponds to 3316–15634 nm of the MIR pulse (wavenumber: 3015–640 cm$^{-1}$). Nonetheless, this wavelength range is sufficient for chemical imaging.

Finally, a silicon-based hyperspectral camera (EBA JAPAN, SIS-H-0.45nm) was used to detect images for each wavelength, the so-called hyperspectral image. For hyperspectral imaging, the data cube consists of 640 × 480 pixel images with 1069 wavelength points. The wavelength resolution of hyperspectral imaging was 0.15 nm, which corresponded to a wavenumber resolution of 2.6–3.7 cm$^{-1}$. It takes a minimum of 8 s to obtain a hyperspectral image data cube. On the other hand, in the case of the discrete frequency imaging, the SFG signal was switched to a high-speed gated image intensifier unit (Hamamatsu, 10880-13F) via mirror M5. A high-speed camera (Phantom, VEO 410L, maximum number of pixels is 1280 × 800) was installed behind the image intensifier unit to measure the amplified image signals. The wavelength was selected by a tunable bandpass

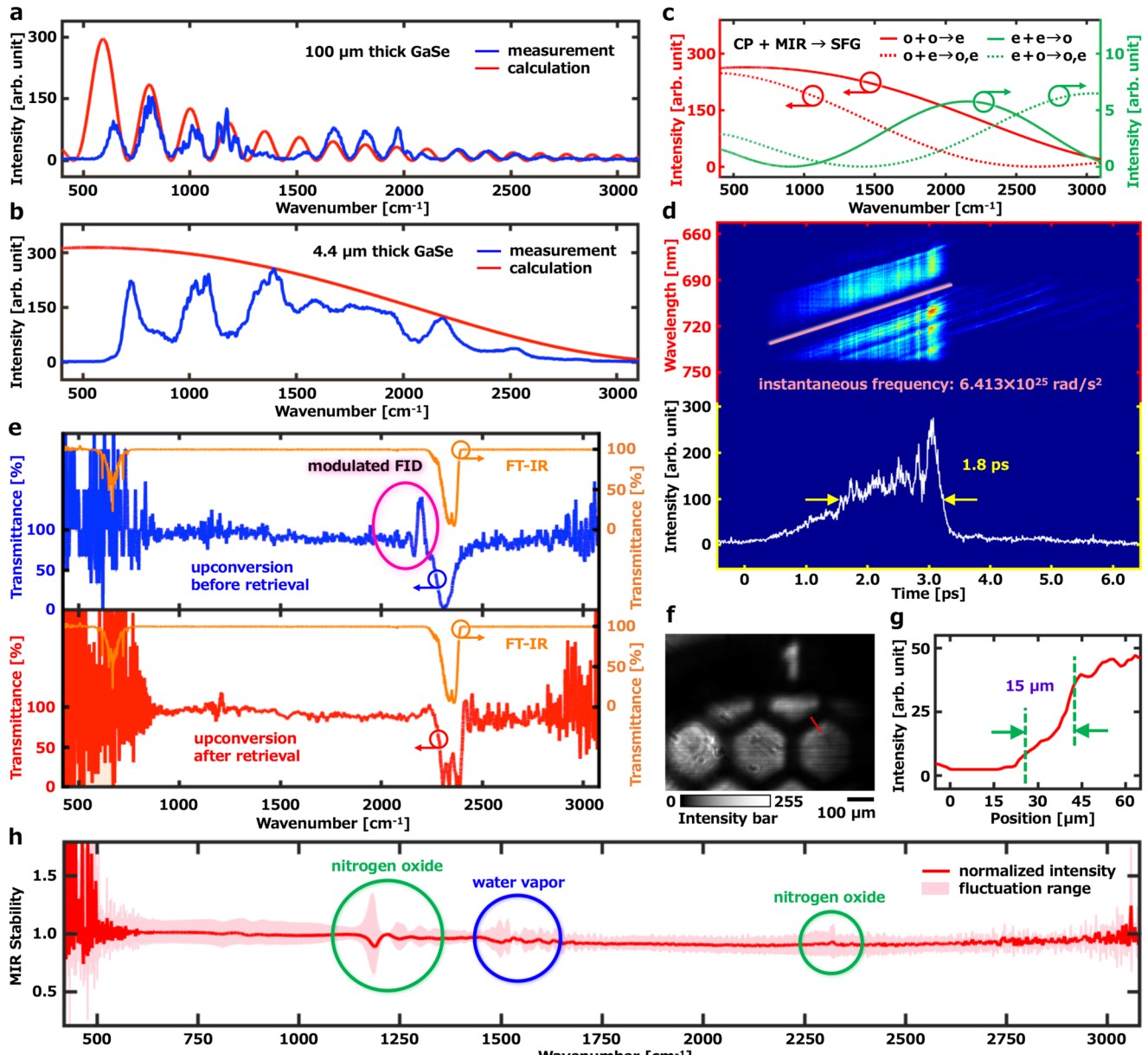

**Fig. 2 | MIR spectrum and spatial resolution of the imaging system.** Upconverted MIR spectrum by **a** 100 μm and **b** 4.4 μm thick GaSe crystal. **c** Calculation results of the configuration of GaSe crystal-type. **d** Spectrally resolved cross-correlation between the MIR pulse and the chirped pulse. **e** Upconversion MIR spectra of $CO_2$ gas before and after retrieval. Infrared transmittance spectrum of $CO_2$ gas by a commercially available FT-IR. **f** Image of a copper mesh grid. **g** Intensity profile marked by the red dashed line in **f**. The spatial resolution is estimated as 15 μm and is defined by the edge response of 10–90% distance of the line profile. **h** MIR hyperspectral stability at an exposure time of 8 s. CP: chirped pulse, MIR: mid-infrared, SFG: sum frequency generation, o: ordinary ray, e: extraordinary ray, FID: free induction decay.

filter set (Semrock TBP01-704/13-25x36 and TBP01-790/12-25x36) before the image intensifier unit. The tunable bandpass filter set can choose any bandwidth of 50 cm⁻¹ within 640–3015 cm⁻¹. The trigger signal of the image intensifier unit and the high-speed camera was synchronized with the laser light source. Discrete frequency imaging achieves a video rate of 5 kHz. The optical configuration shown in Supplementary Information Fig. S1 is in microscope mode with an observable field of view of 800 μm × 600 μm. The fastest measurement time is 8 s for a 640 × 480 pixel image, which is much larger than that obtained by a focal plane array detector (typically 64 × 64 and maximally 128 × 128 pixels) used in an FT-IR microscope. Furthermore, the speed of hyperspectral imaging can be improved by skipping some pixel rows. As shown in Supplementary Information Fig. S1, the maximum speed reaches 0.32 s for a 640 × 20 (skipped every 25 pixel rows)

pixel image and can be used as a rough screening. In subsequent measurements, we use the integration time of 8 s without any pixel rows skips. On the other hand, simply increasing the distance between the sample and the CM4 and adjusting the position of the imaging lens in front of the camera will increase the observable field of view up to 12 mm × 9 mm.

## MIR spectrum and spatial resolution of the imaging system

The optimal crystal thickness must be found to achieve phase matching of 640–3015 cm⁻¹. The upconverted spectrum using thicker crystals (e.g. 100 μm thick GaSe in Fig. 2a) contains fringes, which destroy the wavenumber resolution. As shown in Fig. 2b, c, when the crystal thickness is below 5 μm and the phase matching condition is Type I, namely, sum frequency (extraordinary wave) generation

between the chirped pulse (ordinary wave) and MIR (ordinary wave), the upconversion intensity of 640–3015 cm$^{-1}$ will not go down to zero. A GaSe film with a thickness of less than 5 μm was obtained by the scotch tape method and was attached to a 3 mm thick fused silica substrate (see Supplementary Information Fig. S2). We deduce that the thickness of the GaSe film is about 4.4 μm by comparing the measured spectrum with the calculated one in Fig. 2b.

In order to evaluate the accuracy of the infrared absorption spectrum measurement, the absorption spectrum of $CO_2$ gas was measured. The measured transmission spectrum is shown in the upper row of Fig. 2e. The oscillation which starts from -2400 cm$^{-1}$ to the low-frequency region is due to cross-phase modulation induced by the upconversion with a chirped pulse. This cross-phase modulation is removed by correcting the parabolic phase due to the chirp of the Fourier transform of the measured spectrum, that is, the auto-correlation function of the upconversion electric field[52]. The instantaneous frequency used for the correction is obtained from the spectrally resolved cross-correlation between the chirped pulse and the MIR pulse (Fig. 2d), namely, the SFG spectrum measurement while scanning the delay time between the chirped pulse and the MIR pulse. The lower row of Fig. 2e shows the transmission spectrum of $CO_2$ gas after the correction of the cross-phase modulation. The absorption lines around 2300–2400 cm$^{-1}$ due to $CO_2$ gas were well-reproduced and these are consistent with the FT-IR spectrum measured at a resolution of 1 cm$^{-1}$. The wavenumber resolution of our system is 2.6–3.7 cm$^{-1}$, which is determined by the wavelength resolution (0.15 nm) of the hyperspectral camera. In the following measurements, the wavelength of the retrieved MIR spectrum was calibrated with the $CO_2$ absorption lines. The hyperspectral image data cube at the condition where the whole system is purged with nitrogen (humidity <0.5%RH) is used as a background to calculate the transmittance spectrum of the sample.

The spatial resolution of the imaging system was measured with a copper mesh grid and the width of the bar of the copper mesh grid is 30 μm. Figure 2f, g show a copper mesh grid image and the intensity profile of a bar, respectively. Here, the spatial resolution is estimated to be 15 μm by the intensity profile marked with the red dashed line in Fig. 2f. It is defined by the edge response of 10–90% distance of the line profile.

The bit depth of the MIR hyperspectral chemical imaging is 12 bit and the maximum SNR is 1365. As shown in Fig. 2h, the root mean square of the normalized MIR intensity was 2.77%, while the fluctuations of three absorption bands marked by circles appeared randomly. The two absorption bands (at approximately 1200 cm$^{-1}$ and 2300 cm$^{-1}$) marked by green circles are likely to arise from a complex series of nitrogen oxide formations from the ionization of the filament, and the absorption band marked by the blue circle is due to the water vapor with the ionization and recombination. The absorption bands of nitrogen oxides appeared randomly, and it affected the measurement of the spectrum.

### Identification and mapping of multiple chemical compounds

We tested the analytical capability of our system for chemical imaging by injecting five different kinds of samples (i.e. glycerin, glucose, albumin, 1,2-dioleoyl-sn-glycero-3-phosphocholineglucose and soybean oil) into microfluidic. Figure 3a shows a reflected electron image of a 5 microchannel device that was taken by a scanning electron microscope. The microfluidic device was fabricated from a 20 mm × 20 mm silicon wafer with an overall thickness of 425 μm, a channel width of 100 μm, and a depth of 25 μm. A photograph and design of the microfluidic device are shown in Supplementary Information Figs. S3 and S4.

Here, we prepared 5 samples where glycerin and soybean oil were liquids at room temperature (27 ˚C), glucose was 50 mg/mL aqueous solution, albumin was 10 mg/mL aqueous solution, and 1,2-dioleoyl-sn-glycero-3-phosphocholine was a mixture with water (specific weight of

1,2-dioleoyl-sn-glycero-3-phosphocholine/water was 3/100). 0.5 μL of each was extracted and injected into 5 microchannels respectively. Figure 3b is an optical microscopic image of the visible light region by a white light illumination obtained before the measurement of MIR hyperspectral imaging. The 5 samples were apparently unable to be qualitatively differentiated by the visible light image. After that, the microchannel device was carried to the MIR hyperspectral chemical imaging system. We should note that, since the chamber purging of nitrogen of the MIR hyperspectral chemical imaging system requires 30 min to adequately remove water vapor and carbon dioxide from the air, the water in the glucose and albumin aqueous solution, and 1,2-dioleoyl-sn-glycero-3- phosphocholine/water mixture were evaporated and cast films were formed. The purpose of chemical imaging is to create a visual image of component distribution. Since the spectral signatures of the 5 different kinds of liquids should be different, the hyperspectral image data cube can map the spatial distribution of sample components.

Here, we need to extract the spectra of 5 samples as teaching data for the hyperspectral analysis. We determined the wavenumber that extracts preliminary teaching data based on the known infrared transmittance spectra of that 5 samples (see FT-IR data in Fig. 3d), i.e. peaks 851 cm$^{-1}$ ($CH_2$ rocking vibration), 920 cm$^{-1}$ (C–O symmetric stretching vibration) and 1030 cm$^{-1}$ (C–O stretching vibration) for glycerin, peaks 1142 cm$^{-1}$ ($CH_2$ bending), 1360 cm$^{-1}$ (C–1–H bending) and 1427 cm$^{-1}$ ($CH_2$ bending) for glucose, peaks 1542 cm$^{-1}$ (N–H bending and C–N stretching of Amide II) and 1658 cm$^{-1}$ (C=O stretching vibrations of amide I) for albumin, peaks 1089 cm$^{-1}$ ($PO_2$ symmetrical stretching) and 1250 cm$^{-1}$ ($PO_2$ asymmetrical stretching) for 1,2-dioleoyl-sn-glycero-3-phosphocholine, and peaks 1463 cm$^{-1}$ ($CH_2$ bending vibrations) and 1747 cm$^{-1}$ (CO stretching vibrations) for soybean oil. In addition to the above-mentioned peaks, there are many peaks for choice. Here, we selected the above peaks based on the consideration of not causing spectral interference with other substances. Next, the location information of the microchannels corresponding to each sample was determined by the additive averaging analysis and the peak intensity slope analysis. After that, we randomly selected 20 regions from each microchannel according to the location information of each sample and re-extracted the final teaching data for 5 samples. The hyperspectral image data cube was mapped using the final teaching data by linear discriminant analysis. The mapping results are shown in Fig. 3c, from left to right are glycerin, glucose, albumin, 1,2-dioleoyl-sn-glycero-3-phosphocholine and soybean oil. Figure 3d shows each MIR transmittance spectrum of each sample shown in Fig. 3c. Figure 3e displays images corresponding to fifteen distinct individual wavelength components, while Supplementary Movie 1 demonstrates the continuous scanning of wavelengths across the entire mid-infrared bandwidth. The above five samples include carbohydrates, lipids, and proteins, all of which are common and important substances in biological, food, pharmaceutical, and industrial fields. The test results show that our system is useful in the related fields.

### High-speed imaging of discrete wavelengths

Discrete frequency chemical imaging refers to high-speed imaging of a specific wavenumber. The highest time resolution of the system is evaluated by imaging the fast-flowing water in a microchannel at a fixed frequency band. As shown in Fig. 4a and Supplementary Movie 2, we fixed the wavenumber at 1640 cm$^{-1}$ with a bandwidth of 50 cm$^{-1}$ which is close to an absorption band of water (O–H bending mode at -1640 cm$^{-1}$). The wavenumber was selected by the tunable bandpass filter set before an image intensifier unit. Water was filled in the three microchannels. After ~2 min, the water in the microchannel began to evaporate. The water in the microchannel started to flow in the direction where it did not evaporate due to the surface tension. The water close to the surface of the microchannel wall forms a

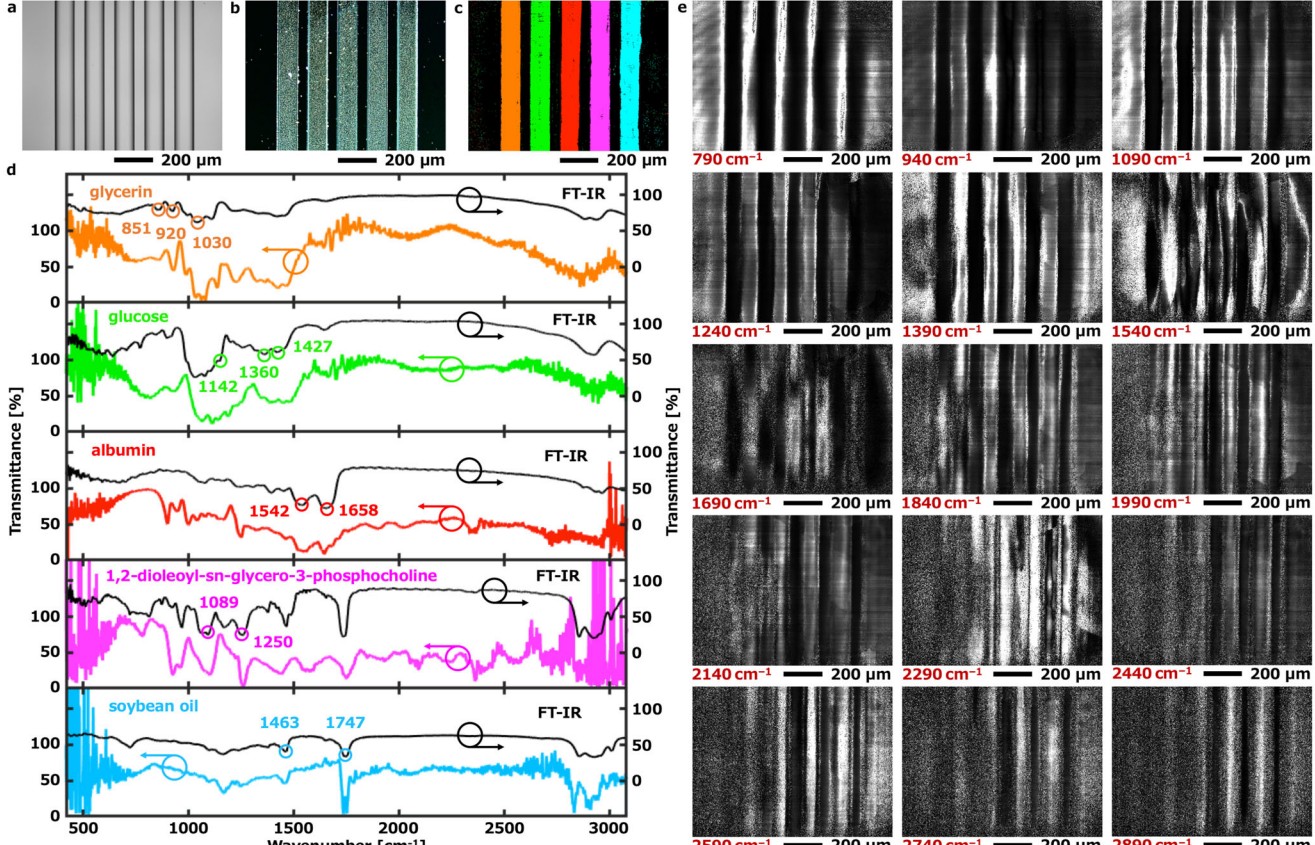

**Fig. 3 | Hyperspectral image for qualitative analysis. a** Reflected electron image of a 5 microchannels device. **b** Optical microscopic image of microchannels device injected with 5 samples obtained before the measurement of mid-infrared hyperspectral imaging. **c** Mapped the mid-infrared hyperspectral image, from left to right are glycerin, glucose, albumin, 1,2-dioleoyl-sn-glycero-3-phosphocholineglucose and soybean oil. The mapped color is a pseudo-color. **d** Each MIR transmittance spectrum of each sample shown in **c**. The black lines are infrared transmittance spectra of the 5 samples measured in a vacuum (4 Pa) by FT-IR (JASCO, FT/IR-6100). **e** Images corresponding to fifteen different individual wavelength components. A movie (Supplementary Movie 1) of the 2D image while the wavelength is continuously scanned over the entire mid-infrared bandwidth.

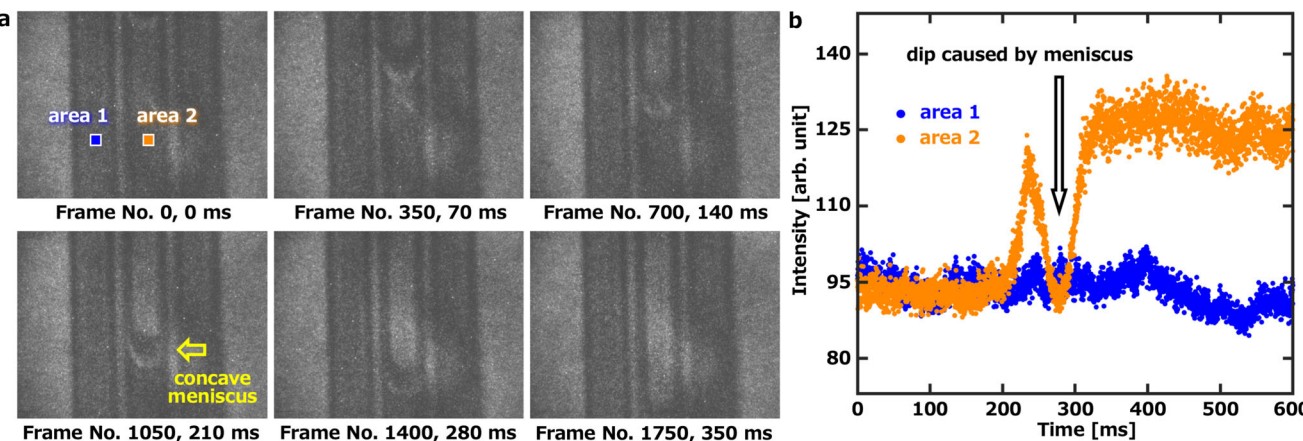

**Fig. 4 | Discrete frequency MIR images of water.** Water was filled into three microchannels with a wide of 200 μm and deep of 25 μm. In the middle microchannel, water starts to evaporate from the upper and the water flows due to surface tension. The water in the left and right microchannels has not yet started to evaporate, thus the MIR has been absorbed all the time and shows a dark distribution. The MIR is no longer absorbed by the water at the position after the water has flowed away, thus showing a brighter distribution. The time resolution is 0.2 ms and the exposure time is 1 μs per frame. **a** 6 frames shown here are images extracted every 350 frames, that is, every 70 ms. The size of the images is 1200 μm × 900 μm (640 × 480 pixels). The bit depth of the high-speed imaging of discrete wavelengths is 8 bit. **b** Temporal evolution of the intensity at the two marked 30 × 30 pixels. The recorded video is given in Supplementary Movie 2.

concave meniscus caused by surface tension. The transmitted MIR light was refracted at the concave meniscus. In Fig. 4b, the dip in time-intensity plots is due to the refraction of the transmitted MIR light by the meniscus. The time resolution was 0.2 ms, which was 100 times faster than the previous QCL-based method[37], and the exposure time was 1 μs per frame. Here, the time resolution was limited by the repetition rate of the laser. A quantitative result can be obtained from the ratio of signal intensities before and after water evaporation. In this measurement, the average flow velocity of water in this microchannel was 2.5 mm/s. This result demonstrates the great potential of our method for fast spatial imaging of rapidly changing water systems.

## MIR hyperspectral imaging and mapping of the onion (A. cepa) bulb leaves epidermal cells

The analytical ability is tested by imaging a more complex composition — A. cepa bulb leaves epidermal cells. Fresh A. cepa bulb leaves epidermal cells were affixed to $CaF_2$ substrates with a thickness of 200 μm. In Fig. 5a, the cell wall, cytoplasm, and some nuclei can be roughly distinguished by morphology. In particular, the distribution of nuclei cannot show contrast in visible light microscopy. The water content of fresh A. cepa bulb leaves epidermal cells is about 90% of the total weight and hence the MIR absorption characteristics of other substances are almost all submerged in the saturated broad spectrum of water. In the experiment, the average thickness of A. cepa bulb leaves epidermal cells was reduced from 110 μm to about 48.8 μm after being

placed in nitrogen gas for about one hour, and the average weight was reduced by about 69.5% because the sample was dried. An MIR hyperspectral image of A. cepa bulb leaves epidermal cells was obtained by removing the background signal of the $CaF_2$ substrate without the sample. Note that any potential effects (e.g. heating) of MIR pulse irradiation were not observed. Even if the sample was placed at the focal point of the MIR beam and irradiated for a few hours, the temperature of the sample hardly changed.

We randomly selected 125 non-overlapping areas of pixel size which limited $4 < x < 35$, $3 < y < 55$ and $20 < x \times y < 720$ in the whole hyperspectral image to construct teaching data and classified them into four classes to map the components. The spectral angle mapper analysis results are shown in Fig. 5b–e, mapped images of the cell wall, cell membrane, nucleus, and cytoplasm were obtained, and their overlap is shown in Fig. 5f. Figure 5g shows the infrared transmittance spectra of the cell wall, cell membrane, cell nucleus, and cytoplasm by extracted hyperspectral image. In the infrared transmittance spectrum of the cell nucleus, characteristic markers of backbone conformation of sugar-phosphate vibrations of DNA were observed, which are, the antisymmetric $PO_2$ stretching band appearing at approximately $1225\,cm^{-1}$, and the symmetric $PO_2$ stretching mode appearing at $1085–1090\,cm^{-1}$. The $1770\,cm^{-1}$ absorption band is assigned to lipids, which may originate from the nuclear membrane.

To verify the accuracy of the mapping results, after the hyperspectral image measurement, the sample was stained with acetocarmine solution and was observed by a visible light microscope. As shown in Fig. 5h, the mapped result of the cell nuclei is similar to the stained result. Carbohydrates were observed in all areas and tended to saturate in the cytoplasm. Proteins (amide I and II) were observed mainly in the cytoplasm and the cell nucleus. The absorption peaks of amide I and II may overlap with the absorption at around $1640\,cm^{-1}$ of some residual water in the cytoplasm leading to the indistinct peaks. Since the thickness of the cell wall was thicker than in other areas, a strong absorption was observed in the entire band in the spectrum of the cell wall. For the cell membrane, the peaks at $1224\,cm^{-1}$ (phosphate asymmetric bands), $1463\,cm^{-1}$ ($CH_2$ bending vibrations), and $1747\,cm^{-1}$ (C=O stretching vibrations) were observed and were attributed primarily to lipids. The distribution of the cell membrane should completely coincide with the protoplasmic layer (including the cytoplasm and nucleus) and is indistinguishable. However, water loss causes the separation of the protoplasmic layer from the cell wall, therefore the distribution of the cell membrane close to the cell wall in Fig. 5c is due to plasmolysis. Therefore, In Fig. 5b, d, e, the constituents other than the cell membrane are the main components, hence they are assigned to cell walls, nucleus, and cytoplasm, respectively. The above results indicate that our method also has great potential for cell analysis.

## MIR hyperspectral imaging and mapping of the section of the mouse embryo

In addition to microscopic observation, the advantage of our system is also significant when being used to screen large areas. The field of view of our system can be up to 12 mm × 9 mm for the large samples. Here, we explain the hyperspectral chemical imaging for a section of the mouse embryo with the millimetric scale view. The sample was a 13.5 days age mouse embryo sagittal plane with a thickness of 3 μm fixed on a 100 μm thick silicon substrate. Figure 6a is a photograph taken with white light illumination. For hyperspectral image, we randomly selected 40 areas of pixel size which limited $6 < x < 115$, $4 < y < 40$ and $40 < x \times y < 1050$ to construct teaching data. As shown in Fig. 6b, two different components were classified and mapped by spectral angle mapper analysis. Figure 6c shows MIR transmittance spectra of two different components which are extracted from the hyperspectral image. Peaks $1542\,cm^{-1}$ (amide II) and $1658\,cm^{-1}$ (amide I) are the characteristic markers for protein. In the mouse embryo, all tissues should contain proteins, while the component concentrated in the diencephalon and

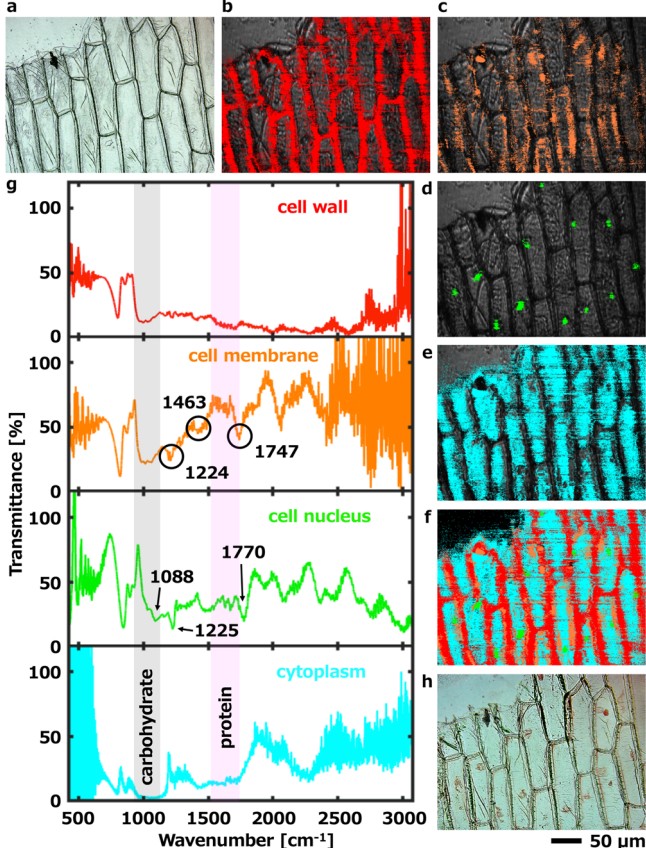

**Fig. 5 | MIR hyperspectral imaging and mapping of the onion (A. cepa) bulb leaves epidermal cells. a** Microscopy image illuminated by visible light. Mapped hyperspectral images of **b** cell wall, **c** cell membrane, **d** cell nucleus, and **e** cytoplasm and **f** their overlap were obtained by spectral analysis. The mapped color is a pseudocolor. **g** Infrared transmittance spectra of the cell wall, cell membrane, cell nucleus, and cytoplasm by extracted hyperspectral image. **h** Stained visible light microscopy image with acetocarmine solution which took after the hyperspectral image measurement. The scale bar applies to all images in Fig. 5.

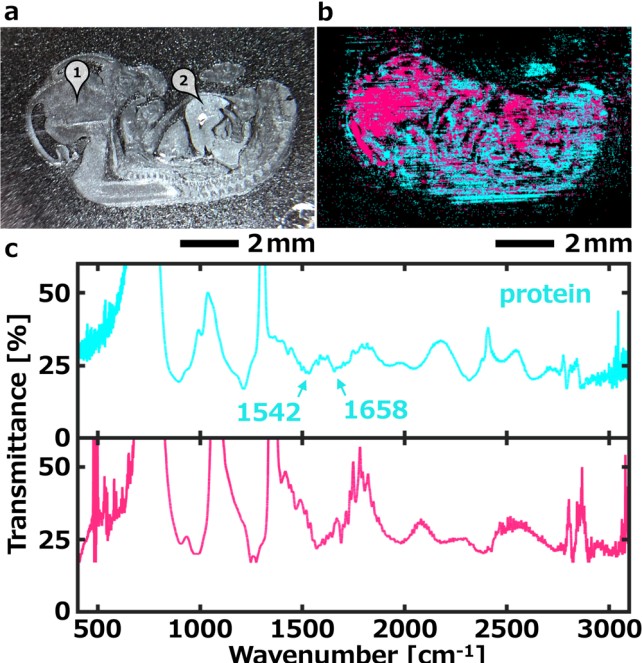

**Fig. 6 | Hyperspectral image of a sagittal plane section of 13.5 days age mouse embryo. a** Photograph by white light illumination. Labels assigned based on anatomy and morphology: (1) diencephalon and (2) muscle. **b,** Mapped the mid-infrared hyperspectral image of 2 classified components. The mapped color is a pseudo-color. **c,** MIR transmittance spectra of 2 classes which are extracted from hyperspectral image.

liver is unknown, most likely lipids. While no detailed analysis of the underlying biochemistry was performed, the classified 2 classes are highly consistent with anatomical- and morphological-based judgments. The classification here depends on the scale and the granularity of the desired observation. More detailed classes can be subdivided by increasing the correlation between spectra in hyperspectral image data cube analysis. Admittedly, human factors were involved in the classification process. Because of this, in future hyperspectral image analysis, machine learning and artificial intelligence should be introduced to assist in classification.

For the section of the mouse embryo, our system only needs 8 s to obtain the hyperspectral chemical imaging of 1069 point wavenumbers, which covered 640–3015 cm⁻¹. Whereas for this task, the imaging of a ~10 mm sample would take tens of hours using a QCL-based discrete frequency imaging system and would take an FT-IR system upward of 1000 h or more. In contrast, our results indicate that our method has great potential not only for microscopic scale analysis, such as cells, but also for centimeter scale analysis, such as biological tissues (e.g., surgical tissue sections, etc.).

## Discussion

We implemented high-speed scanless MIR hyperspectral chemical imaging using sum frequency generation for upconversion by a broad-spectrum MIR ultrashort pulse as the light source. The light source of the broad-spectrum MIR ultrashort pulse saved the time required for wavelength scanning. The up-converted visible light can be imaged with a silicon-based detector while maintaining the infrared absorption spectrum. The SNR is much higher than that of the traditional MIR detector, which greatly improves the sensitivity. The observable field of view is from the micro- to the centimeter scale. The measurable wavenumber band is 640–3015 cm⁻¹ and an MIR hyperspectral image of 640 × 480 pixels with 1069 wavenumber points can be obtained in the shortest time of 8 s. Furthermore, the maximum speed of

hyperspectral imaging reaches 0.32 s for a 640 × 480 (skipped every 25 pixel rows) pixel image, and it can be applied for rough screening. The measurement speed for discrete frequency imaging reaches a video rate of 5 kHz. The chemical imaging results of the microfluidic device, onion (*A. cepa*) bulb leaves epidermal cells, and sections of mouse embryos fully confirm the validity of our method and its great potential in chemical imaging. We believe that the method of this study has broad application prospects in many fields such as chemical analysis, life sciences, and medicine.

## Methods

### Manufacture of the microfluidic device

The microfluidic device was fabricated by performing the etching process on the silicon wafer. The following is the manufacturing process.

1. Resist coating
   A positive photoresist was coated to a silicon wafer (thickness 525 μm) with a thickness of 1.4 μm. After that, the solvent was removed by heating at 90 °C for 2 min.
2. Exposure
   The resist-coated silicon wafer was irradiated with ultraviolet light of 370–380 nm wavelength using a maskless exposure device (Japan Science Engineering Co., Ltd., MX-1204). The exposure of 150 mJ/cm² modified the pattern of the microfluidic part into alkali-soluble. The computer-aided design (CAD) of the microfluidic device is shown in Supplementary Information Fig. S3.
3. Removal of resist
   The portion of the alkali-soluble modified resist was removed by immersing in a tetramethylammonium hydroxide developer for 90 s. After that, the remaining developer and residue were washed with pure running water for 2 min.
4. Fixing the photoresist
   The photoresist was baked and hardened by heating at 120 °C for 2 min.
5. Etching
   The silicon in the portion from which the photoresist was removed was deep-drilled etched by the Bosch process, which is by repeating the three steps of isotropic etching of Si with $SF_6$ gas, deposition of protective film with $C_4F_8$ gas, and anisotropic etching (removal of protective film on the bottom surface) of Si with $SF_6$ gas.
6. Peeling of resist
   The photoresist was dissolved and removed by hot dipping the wafer for 10 min in a mixed solution of sulfuric acid and hydrogen peroxide at a liquid temperature of 110–130 °C.
7. Cleaning after etching
   The wafer was washed with pure running water for 2 min to remove the remaining mixed solution and residue.
8. Dicing
   20 mm × 20 mm microfluidic device chips were cut using a dicing saw as shown in the design drawing.

### Reagent list

- Glycerin
  Glycerol, CAS RN: 56-81-5, Hayashi Pure Chemical Ind., Ltd.
- Albumin
  Albumin, from Egg, CAS RN: 9006-59-1, FUJIFILM Wako Pure Chemical Corporation.
- Glucose
  D(+)-glucose, CAS RN: 50-99-7, FUJIFILM Wako Pure Chemical Corporation.
- Soybean oil
  Soybean Oil, CAS RN: 8001-22-7, FUJIFILM Wako Pure Chemical Corporation.

- 1,2-dioleoyl-sn-glycero-3-phosphocholine
  1,2-dioleoyl-sn-glycero-3-phosphocholine, CAS RN: 4235-95-4, Tokyo Chemical Industry Co., Ltd.

## Hyperspectral image data cube analysis

- Additive averaging analysis
  The additive averaging analysis is based on the average spectral intensity of any wavelength region. By setting the wavelength range and threshold (intensity of maximum and minimum), the intensity distribution in the specified wavelength range within the threshold range is mapped.
  The addition average value $I_{ave}$ is calculated by

$$I_{ave} = S/n \qquad (1)$$

  where $S$ is the summation of the intensities of each pixel at the selected wavelength, $n$ is the number of wavelength points.
- Peak intensity slope analysis
  The peak intensity slope analysis is based on the difference in spectral intensity between the two selected wavelengths. The spectral intensity difference is analyzed using the normality differential spectral index (NDSI) and is calculated by

$$NDSI = (I_{\lambda 1} - I_{\lambda 2})/(I_{\lambda 1} + I_{\lambda 2}) \qquad (2)$$

  where, $I_{\lambda 1}$ and $I_{\lambda 2}$ are the intensities of the two selected wavelengths, respectively. By specifying the threshold range of NDSI, the component of the specified spectral intensity difference is mapped.
  We should note that, when the MIR absorption bands of different samples overlap, the additive averaging method may not be applicable, but usually organic compounds have more than two infrared absorption peaks and are suitable for the peak intensity slope method. In the analysis of identification and mapping of multiple chemical compounds of the microfluidic device, peaks that do not cause spectral interference with other species were selected for analysis.
- Spectral angle mapper analysis
  The spectral angle mapper analysis classifies each pixel into the closest class of the teaching spectra. The correlation between each pixel and each teaching spectrum is calculated by the inner product of each pixel spectrum vector ($I_{\lambda 1}$, $I_{\lambda 2}$, ..., $I_{\lambda n}$) and the teaching spectrum vector ($I_{t\lambda 1}$, $I_{t\lambda 2}$, ..., $I_{t\lambda n}$). Each pixel is classified into a class of teaching spectra with the maximum internal product value and mapped.
- Linear discriminant analysis

The linear discriminant analysis classifies pixels that have the same or similar spectrum as the teaching spectrum based on Fisher's linear discriminant. It is well known that Fisher's linear discrimination is to find a projection vector $\boldsymbol{w}$ that maximizes the degree of separation (Fisher criterion $J(\boldsymbol{w})$) of the projected teaching spectrum class. Here, the Fisher criterion $J(\boldsymbol{w})$ is defined as

$$J(\boldsymbol{w}) = \frac{\boldsymbol{w}^T \boldsymbol{S_B} \boldsymbol{w}}{\boldsymbol{w}^T \boldsymbol{S_W} \boldsymbol{w}} \qquad (3)$$

where, $\boldsymbol{S_B}$ is the between-class variance and $\boldsymbol{S_W}$ is the within-class variance. The attribution of the pixel was determined according to the projection vector $\boldsymbol{w}$.

## Handling of onion (*A. cepa*) bulb leaves epidermal cells
A bulb of onion (*A. cepa*) was divided into 8 equal parts with a sagittal plane, and one bulb leaves was picked off. A scalpel cut was made near the center of the bulb leaves in a grid pattern of about 10 mm × 10 mm. The 10 mm × 10 mm epidermis was peeled off with tweezers and placed on a 200 μm thick, 20 mm × 20 mm CaF₂ substrate. The epidermis was dripped with a drop of water, and the four corners of the epidermis were pulled with a dissecting needle to spread it flat. Excess water was sucked off with filter paper.

## Section of the mouse embryo
The sample was a 13.5 days age mouse embryo sagittal plane with a thickness of 3 μm fixed on a 100 μm thick silicon substrate and was purchased from GenoStaff Co., Ltd.

## Statistics and reproducibility

- The experiment of Fig. 2a, b was repeated 58 times independently with similar results.
- The experiment of Fig. 2d was repeated 128 times independently with similar results.
- The experiment of Fig. 2e was repeated 24 times independently with similar results.
- The experiment of Fig. 2f was repeated 67 times independently with similar results.
- The experiment of Fig. 2h was repeated 38 times independently with similar results.
- The experiment of Fig. 3 was repeated 56 times independently with similar results.
- The experiment of Fig. 4 was repeated 7 times independently with similar results.
- The experiment of Fig. 5 was repeated 8 times independently with similar results.
- The experiment of Fig. 6 was repeated 7 times independently with similar results.

## Reporting summary
Further information on research design is available in the Nature Portfolio Reporting Summary linked to this article.

## Data availability
All data generated in this study have been deposited in the figshare database under accession code https://doi.org/10.6084/m9.figshare.22849559.

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

## Acknowledgements

This study was supported by the Japan Science and Technology Agency (JST) Core Research for Evolutional Science and Technology (CREST), Grant Number JPMJCR17N5, T.F. This work was supported by Frontier Photonic Sciences Project of National Institutes of Natural Sciences (NINS) (Grant Number 01212208), Y.Z. The microfabrication of micro-fluidic device of this work was supported by "Nanotechnology Platform Japan" (Grant Number JPMXP09F21TT0043) and "Advanced Research Infrastructure for Materials and Nanotechnology (ARIM)" (Grant Number JPMXP1222TT0006) of the Ministry of Education, Culture, Sports, Science and Technology (MEXT), Japan, Y.Z. and T.F.

## Author contributions

T.F. conceived the concept and supervised the work. Y.Z. and T.F. conceived and designed the experimental setup. Y.Z. built the experimental setup. Y.Z. conceived and prepared the sample. Y.Z., S.K., W.H., and T.F. performed the experiment. Y.Z., S.K., and T.F. analyzed the data. Y.Z., Y.F., and T.F. analyzed the infrared spectrum. Y.Z. and T.F. wrote the manuscript. Y.Z., S.K., T.F., W.H., C.L., and T.F. contributed to the discussion of the results and the manuscript.

## Competing interests

The authors declare no competing interests.

## Ethics approval

Patent applicant: School corporation Toyota Gakuen. Name of inventors: T.F., and Y.Z. Application number: 2021-211167. Status of application: pending. Specific aspect of manuscript covered in patent application: The mid-infrared hyperspectral imaging method in this manuscript is patent pending.
