## [Peer Review File · Nature Communications]

High-speed scanless entire bandwidth mid-infrared chemical imagingREVIEWER COMMENTS

Reviewer #1 (Remarks to the Author):

This manuscript deals with hyperspectral imaging using mid-infrared femtosecond pulses delivered by four-wave difference frequency generation in air. Chirped pulse up-conversion enables the use of more efficient hyperspectral cameras in the visible domain, and also helps in managing the huge bandwidth (3-30 μm) of the incoming mid-infrared pulses. The experimental results are truly impressive and show a great improvement with respect to previous work. There are major implications in terms of chemical imaging, so that this work is clearly of great interest and of a timely topic which could be suitable for publications in Nature Communications. However, some revisions must be first considered to improve the quality of the manuscript.

1) The data shown in the figures do not really do justice to the potential of hyperspectral imaging. I don't think this has anything to do with the experimental measurements, but merely to the way the data are presented in the manuscript. As explained by the authors in the introduction, the difference between multispectral and hyperspectral imaging stems with the number of available frequency channels. The authors claim to have measured 1069 wavelength components, yet no figure shows the actual cube of data mentioned in the manuscript. Only 1D or 2D projections of the cube are shown, even though admittedly the 2D images shown in figures 5 and 6 are processed from the hyperspectral data. But eventually only a few bands are shown, which relates more to multispectral than to hyperspectral imaging. I suggest that the authors show a set of images corresponding to at least ten different individual wavelength components, and also show a movie of the 2D image while the wavelength is continuously scanned over the entire mid-infrared bandwidth. Only then will the reader be able to judge the quality of the hyperspectral data.

2) Figure 3d shows a comparison between the spectra retrieved by up-conversion and FTIR data. Although there is a general qualitative agreement, there are some discrepancies which should be discussed. To give just two examples, the up-conversion data show some wiggles just below 1000 cm^{-1} in albumin and a strong absorption band at 900 cm^{-1} in glucose, two features that are not observed with FTIR. Conversely, FTIR spectra show a strong CO_2 absorption in the 2300-2400 cm^{-1} band, implying that purging was not implemented for the FTIR measurement – a fact that is not mentioned in the manuscript.

3) The plot in Fig. 3c is not really quantitative. It would be useful to have a section for each of the five mapping signals so as to evaluate the amount of crosstalk. For example, when looking at the glycerin mapping signal, what is the relative signal obtained in the other four micro-channels where there is no glycerin ?

4) The quality of the experimental data results for a significant part from the use of chirped pulse up-conversion, yet no references are given to previous applications of this technique in spectroscopy (except for ref. 48 when dealing with cross phase modulation). I would suggest that the authors additionally cite their own work, Nomura et al., Opt. Express 21, 18249 (2013), and also earlier work such as Kubarych et al., Opt. Lett. 30, 1228 (2005), Baiz and Kubarych, Opt. Lett. 36, 187 (2011) and Zhu et al., Opt. Express 20, 10562 (2012).

Reviewer #2 (Remarks to the Author):

In their paper entitled "High-speed full-field entire bandwidth mid-infrared chemical imaging", the authors present broadband (640–3015 cm^{-1}) single-shot upconversion hyperspectral imaging with detection in the near-infrared spectral range. The work is of high quality, the authors claim to present a couple of innovations, and the results are convincing in terms of performance. Given the strong interest of a wide community in modern, coherent IR technologies, Nature Communications seems an appropriate dissemination medium for these results. However, I have a few concerns before deeming

the manuscript suitable for publication:

1. Comparisons with the state of the art, in particular QCL-based IR imaging, are given throughout the text. However, for a fair, overall comparison, an overview table with the relevant parameters (such as spectral coverage and resolution, spatial resolution, SNR, measurement time,....) would be very helpful. This will show the absolute improvement over existing techniques (e.g., those in the Bhargava group).
2. While the manuscript is, in general well-written, there are a few formulations which should be revised, e.g., lines 19, 89/90, 408.
3. Some members of the scientific community have an issue with the wording "half-cycle pulses". While it is debatable whether this is physically correct or not, avoiding this formulation would avoid this debate.
4. The formulation "full-field" is misleading because the method does not measure the full electric field. It should be removed or replaced.
5. Line 43: the claim "identifies chemical composition" might be too strong because of the many lines overlapping in the condensed phase.

Response to the reviewers' comments

Manuscript ID: NCOMMS-22-37327

Title: High-speed scanless entire bandwidth mid-infrared chemical imaging

We thank the reviewers for their very careful reading and kind comments. The following is the response to the reviewers' comments. Reviewers' comments are in black font and our answers are in blue font. The red-lined manuscript is attached at the end of the text.

Reviewer #1 (Remarks to the Author):

This manuscript deals with hyperspectral imaging using mid-infrared femtosecond pulses delivered by four-wave difference frequency generation in air. Chirped pulse up-conversion enables the use of more efficient hyperspectral cameras in the visible domain, and also helps in managing the huge bandwidth (3-30 μm) of the incoming mid-infrared pulses. The experimental results are truly impressive and show a great improvement with respect to previous work. There are major implications in terms of chemical imaging, so that this work is clearly of great interest and of a timely topic which could be suitable for publications in Nature Communications. However, some revisions must be first considered to improve the quality of the manuscript.

1) The data shown in the figures do not really do justice to the potential of hyperspectral imaging. I don't think this has anything to do with the experimental measurements, but merely to the way the data are presented in the manuscript. As explained by the authors in the introduction, the difference between multispectral and hyperspectral imaging stems with the number of available frequency channels. The authors claim to have measured 1069 wavelength components, yet no figure shows the actual cube of data mentioned in the manuscript. Only 1D or 2D projections of the cube are shown, even though admittedly the 2D images shown in figures 5 and 6 are processed from the hyperspectral data. But eventually only a few bands are shown, which relates more to multispectral than to hyperspectral imaging. I suggest that the authors show a set of images corresponding to at least ten different individual wavelength components, and also show a movie of the 2D image while the wavelength is continuously scanned over the entire mid-infrared bandwidth. Only then will the reader be able to judge the quality of the hyperspectral data.

Authors' answer: We added a set of images corresponding to fifteen different individual wavelength components in Fig. 3 and also added a movie of the 2D image while the wavelength is continuously scanned over the entire mid-infrared bandwidth.

We added the following description in the caption of Fig. 3.

“e, Images corresponding to fifteen different individual wavelength components. A video (Supplementary Video 1) of the 2D image while the wavelength is continuously scanned over the entire mid-infrared bandwidth.”

The Supplementary Video 1 of the previous version has been renumbered to Supplementary Video 2 in this version.

2) Figure 3d shows a comparison between the spectra retrieved by up-conversion and FTIR data. Although there is a general qualitative agreement, there are some discrepancies which should be discussed. To give just two examples, the up-conversion data show some wiggles just below 1000 cm⁻¹ in albumin and a strong absorption band at 900 cm⁻¹ in glucose, two features that are not observed with FTIR. Conversely, FTIR spectra show a strong CO₂ absorption in the 2300-2400 cm⁻¹ band, implying that purging was not implemented for the FTIR measurement – a fact that is not mentioned in the manuscript.

Authors' answer: The measurement method employed in this study is susceptible to the effects of refractive index inhomogeneity, which can manifest as an optical path difference in the transmitted MIR light. To mitigate this effect, we endeavored to maintain a uniform sample thickness during preparation. Nonetheless, variations in thickness or concentration along the same flow path can introduce observable contrasts, as depicted in Fig. 3e and Supplementary Video 1. Notably, in the case of teacher data selection, significant differences in thickness or concentration within the selected area can give rise to some wiggles in the up-conversion spectrum after retrieval. For example, the up-conversion spectrum of albumin exhibits discernible "wiggles" at frequencies just below 1000 cm⁻¹ and 1250 cm⁻¹, which are not apparent in FT-IR spectra.

In the revised manuscript, we have substituted the earlier Fig. 3 with the latest measurement results. Notably, the updated data indicate an improvement in the aforementioned issues.

Furthermore, the FT-IR measurements were conducted under vacuum conditions (4 Pa). This information was added to the Fig. 3 caption.

3) The plot in Fig. 3c is not really quantitative. It would be useful to have a section for each of the five mapping signals so as to evaluate the amount of crosstalk. For example, when looking at the glycerin mapping signal, what is the relative signal obtained in the other four micro-channels where there is no glycerin ?

Authors' answer: Yes, the plot depicted in Figure 3c is primarily qualitative rather than quantitative. As we previously indicated in our response to question 2). The inhomogeneity of the sample's refractive index can induce variations in the optical path difference of the transmitted MIR light, thereby introducing differences in signal intensity at different thicknesses or concentrations of the same sample. However, such intensity differences will be disregarded in spectral angle analysis, which classifies a spectrum based on its proximity to multiple registered teacher data. Nevertheless, it is worth noting that spectral angle analysis suffices for qualitative analysis.

As you pointed out, it would be informative to assess the extent of crosstalk for each of the five mapped signals. Nonetheless, this would require a certain concentration of standard samples to correct. We intend to pursue a more quantitative analysis in future studies.

4) The quality of the experimental data results for a significant part from the use of chirped pulse up-conversion, yet no references are given to previous applications of this technique in spectroscopy (except for ref. 48 when dealing with cross phase modulation). I would suggest that the authors additionally cite their own work, Nomura et al., *Opt. Express* **21**, 18249 (2013), and also earlier work such as Kubarych et al., *Opt. Lett.* **30**, 1228 (2005), Baiz and Kubarych, *Opt. Lett.* **36**, 187 (2011) and Zhu et al., *Opt. Express* **20**, 10562 (2012).

Authors' answer: We have added it in the text as described below.

A general method for characterizing ultrashort mid-infrared pulses is through upconversion using a stretched chirped pulse⁴⁷⁻⁵⁰. (Line 94-95)

47. Kubarych, K. J., Joffre, M., Moore, A., Belabas, N. & Jonas, D. M. Mid-infrared electric field characterization using a visible charge-coupled-device-based spectrometer. *Opt. Lett.* **30**, 1228–1230 (2005).
48. Baiz, C. R. & Kubarych, K. J. Ultrabroadband detection of a mid-IR continuum by chirped-pulse upconversion. *Opt. Lett.* **36**, 187 (2011).
49. Zhu, J., Mathes, T., Stahl, A. D., Kennis, J. T. M. & Groot, M. L. Ultrafast mid-infrared spectroscopy by chirped pulse upconversion in 1800-1000cm⁻¹ region. *Opt. Express* **20**, 10562 (2012).
50. Nomura, Y. *et al.* Single-shot detection of mid-infrared spectra by chirped-pulse upconversion with four-wave difference frequency generation in gases. *Opt. Express* **21**, 18249–18254 (2013).

Reviewer #2 (Remarks to the Author):

In their paper entitled "High-speed full-field entire bandwidth mid-infrared chemical imaging", the authors present broadband (640–3015 cm^{-1}) single-shot upconversion hyperspectral imaging with detection in the near-infrared spectral range. The work is of high quality, the authors claim to present a couple of innovations, and the results are convincing in terms of performance. Given the strong interest of a wide community in modern, coherent IR technologies, Nature Communications seems an appropriate dissemination medium for these results. However, I have a few concerns before deeming the manuscript suitable for publication:

1. Comparisons with the state of the art, in particular QCL-based IR imaging, are given throughout the text. However, for a fair, overall comparison, an overview table with the relevant parameters (such as spectral coverage and resolution, spatial resolution, SNR, measurement time,....) would be very helpful. This will show the absolute improvement over existing techniques (e.g., those in the Bhargava group).

Authors' answer: The suggestion of the reviewer regarding the inclusion of an overview table with relevant parameters to facilitate a comprehensive comparison is appreciated.

However, a straightforward comparison is not feasible due to differences in light source intensity, imaging techniques, optical setups, field of view scales, and spectral region mismatches among different imaging systems.

For instance, the definition of measurement time varies depending on the imaging method used. For example, when using focal plane array (FPA) imaging, images with a larger field of view are usually obtained by stitching mosaics. In addition, discrete frequency imaging involves selecting a specific number of frequency points to form the "data cube," resulting in differences in imaging time. Therefore, it is challenging to generalize the imaging time. A comprehensive and fair comparison requires simultaneous normalization of multiple parameters. However, there is currently no fixed standard for this normalization. A possible criterion for a fair and comprehensive comparison is to measure the same sample using different methods and compare the spectral coverage, wavenumber resolution, spatial resolution, signal-to-noise ratio, and measurement time under the same field of view. However, this requires the cooperation of each group, and the feasibility of this approach is currently limited.

Nonetheless, an overview table is provided solely for reviewers' reference.

2) Figure 3d shows a comparison between the spectra retrieved by up-conversion and FTIR data. Although there is a general qualitative agreement, there are some discrepancies which should be discussed. To give just two examples, the up-conversion data show some wiggles just below 1000 cm⁻¹ in albumin and a strong absorption band at 900 cm⁻¹ in glucose, two features that are not observed with FTIR. Conversely, FTIR spectra show a strong CO₂ absorption in the 2300-2400 cm⁻¹ band, implying that purging was not implemented for the FTIR measurement – a fact that is not mentioned in the manuscript.

Authors' answer: The measurement method employed in this study is susceptible to the effects of refractive index inhomogeneity, which can manifest as an optical path difference in the transmitted MIR light. To mitigate this effect, we endeavored to maintain a uniform sample thickness during preparation. Nonetheless, variations in thickness or concentration along the same flow path can introduce observable contrasts, as depicted in Fig. 3e and Supplementary Video 1. Notably, in the case of teacher data selection, significant differences in thickness or concentration within the selected area can give rise to some wiggles in the up-conversion spectrum after retrieval. For example, the up-conversion spectrum of albumin exhibits discernible "wiggles" at frequencies just below 1000 cm⁻¹ and 1250 cm⁻¹, which are not apparent in FT-IR spectra.

In the revised manuscript, we have substituted the earlier Fig. 3 with the latest measurement results. Notably, the updated data indicate an improvement in the aforementioned issues.

Furthermore, the FT-IR measurements were conducted under vacuum conditions (4 Pa). This information was added to the Fig. 3 caption.

3) The plot in Fig. 3c is not really quantitative. It would be useful to have a section for each of the five mapping signals so as to evaluate the amount of crosstalk. For example, when looking at the glycerin mapping signal, what is the relative signal obtained in the other four micro-channels where there is no glycerin ?

Authors' answer: Yes, the plot depicted in Figure 3c is primarily qualitative rather than quantitative. As we previously indicated in our response to question 2). The inhomogeneity of the sample's refractive index can induce variations in the optical path difference of the transmitted MIR light, thereby introducing differences in signal intensity at different thicknesses or concentrations of the same sample. However, such intensity differences will be disregarded in spectral angle analysis, which classifies a spectrum based on its proximity to multiple registered teacher data. Nevertheless, it is worth noting that spectral angle analysis suffices for qualitative analysis.

As you pointed out, it would be informative to assess the extent of crosstalk for each of the five mapped signals. Nonetheless, this would require a certain concentration of standard samples to correct. We intend to pursue a more quantitative analysis in future studies.

Name & Year	Wavenumber range [cm ⁻¹]	Bandwidth [cm ⁻¹]	Number of measured wavelength	Wavenumber resolution [cm ⁻¹]	pixels	Spatial resolution [μm]	Measurement time [s]	SNR	Method	References URL
this study	640-3015	2375	1069 discrete frequency	2.6–3.7 -	640 × 480 640 × 480	15	8 0.0002	1365 depends on ICCD gain	MIR ultra-short pulse chirped pulse upconversion MIR ultra-short pulse chirped pulse upconversion	
Haase, 2015	1160-1320	160	160	1	640 × 480	9	11.3	15.8-18.2	QCL, microbolometer array (Infatec GmbH, Germany)	https://doi.org/10.1002/jbio.201500264
N. Kröger, 2014	1030-1090 1160-1320	60 160	60 160	1	640 × 480	9.4	52	-	QCL, microbolometer array (Infatec GmbH, Germany)	https://doi.org/10.1117/12.2041988
Bassan, 2016	900-1800	900	1	-	128 × 128	5–10	540	-	Spero (Daylight Solutions Inc., San Diego, CA, USA) QCL, mercury–cadmium–telluride (MCT) detector	https://doi.org/10.1039/C4AN00638K
Bird, 2017	900-1800	900	226	4 or 8	480 × 480	4.38	300	-	Spero (Daylight Solutions Inc., San Diego, CA, USA) QCL, 480 × 480 microbolometer VO ₂ Focal Plane Array (FPA)	https://doi.org/10.1039/C6AN01916A
Kröger-Lui, 2015	1027-1087 1167-1319	60 152	15 38	4	640 × 480	9.4	450	-	QCL, microbolometer array (Infatec GmbH, Germany)	https://doi.org/10.1039/C4AN02001D
Yeh, 2016	777-1940	1163	1	-	500 × 500	7.8	62.5	3000	QCL-MCT, raster scan [0.25 (ms/point)/wavenumber]	https://doi.org/10.1117/12.2230003
Koziol, 2019	1176-1800	624	312	2	480 × 480	-	300	-	QCL, Microbolometer Focal Plane Array (FPA)	https://doi.org/10.1016/j.aca.2018.11.032
Shi, 2020	1200-1800 2000-2300	600 300	150 75	8	480 × 480	-	120 180	-	QCL, Microbolometer Focal Plane Array (FPA), DF-IR	https://doi.org/10.1038/s41592-020-0883-z
Amrania, 2011	1162-1562 2020-2500	400 480	20 24	20	320 × 240	-	0.1	-	OPG, wide spectral response IR camera (CEDIP JADE), MCT-FPA	https://doi.org/10.1039/C0SC00409J
Junaid, 2018	1250-1666	416	1	0.8	4400	35	0.01	-	OPA, SFG by AgGaS ₂ (AGS)	https://doi.org/10.1364/OE.26.002203
Junaid, 2019	2500-4348	1848	62	5.5	64000	35	0.155	-	SFG [1064 nm pump + OPO (20 ps)], lithium niobate crystal	https://doi.org/10.1364/OPTICA.6.000702
Yeh, 2019	770-1940	1170	30	4	20000 × 20000	-	3240000	-	QCL-MCT	https://doi.org/10.1021/acs.analchem.8b04749
Yeh, 2015	776.9-1904.4	1127.5	282	4	128 × 128	17.5	3	260-1034	QCL, focal plane array (FPA) (Lockheed Martin Corporation, Santa Barbara Focal Plane, USA)	https://pubs.acs.org/doi/10.1021/ac5027513

2. While the manuscript is, in general well-written, there are a few formulations which should be revised, e.g., lines 19, 89/90, 408.

Authors' answer: The corresponding formulations have been corrected as follows. (Note: some line numbers are shifted in the revised version.)

Line 20-26: Here we report a mid-infrared hyperspectral chemical imaging technique that uses chirped pulse upconversion of sub-cycle pulses at the image plane. This technique offers a lateral resolution of 15 μm , and the field of view is adjustable between 800 $\mu\text{m} \times 600 \mu\text{m}$ to 12 mm \times 9 mm. The hyperspectral imaging produces a 640 \times 480 pixel image in 8 s, which covers a spectral range of 640–3015 cm^{-1} , comprising 1069 wavelength points and offering a wavenumber resolution of 2.6–3.7 cm^{-1} .

Line 89-93: Here, we propose the most efficient mid-infrared hyperspectral chemical imaging method to our knowledge, which employs scanless global imaging to save time on raster scans. Additionally, it covers the entire MIR bandwidth, and we can create either a monochromatic or a hyperspectral image by using a wavelength filter or a grating to save time on tuning wavelengths.

Line 95-98: We used self-developed, full-spectrum (bandwidth: 3–30 μm), sub-cycle MIR pulses as the light source and performed global imaging, which can produce an image without the need for sample scanning.

Line 415-417: In contrast, our results indicate that our method has great potential not only for microscopic scale analysis, such as cells, but also for centimeter scale analysis, such as biological tissues (e.g., surgical tissue sections, etc).

3. Some members of the scientific community have an issue with the wording "half-cycle pulses". While it is debatable whether this is physically correct or not, avoiding this formulation would avoid this debate.

Authors' answer: We changed the formulation "half-cycle pulses" to "sub-cycle pulses". (Line 21, 96, 114, and 125)

4. The formulation "full-field" is misleading because the method does not measure the full electric field. It should be removed or replaced.

Authors' answer: The term "full-field" was replaced with "scanless" (title, Line 55, 90, and 421, Fig. 1 caption) and "full-field irradiation" was replaced with "global irradiation". (Line 113)

5. Line 43: the claim "identifies chemical composition" might be too strong because of the many lines overlapping in the condensed phase.

Authors' answer: We have modified the corresponding formulation as follows.

"The mid-infrared (MIR) hyperspectral technique enables "chemical imaging" or "chemical mapping" by providing information on the chemical composition of an object through molecular vibrations." (Line 45-47)

Other fixes

[1] A current address has been added because author Yue Zhao has transferred.

[2] We added a fund information.

REVIEWERS' COMMENTS

Reviewer #1 (Remarks to the Author):

The revised manuscript addresses all points raised by both reviewers in a satisfactory manner. I therefore recommend the manuscript to be published in its present form.

Reviewer #2 (Remarks to the Author):

In their revised version, the authors have suitably addressed all of my concerns, as well as those of the other reviewer. I therefore recommend the paper for publication in Nature Communications.